# Borane catalysed ring opening and closing cascades of furans leading to silicon functionalized synthetic intermediates

Chinmoy K. Hazra[1,2], Narasimhulu Gandhamsetty[1,2], Sehoon Park[1,2] & Sukbok Chang[1,2]

The conversion of renewable biomass resources to synthetically valuable chemicals is highly desirable, but remains a formidable challenge in regards to the substrate scope and reaction conditions. Here we present the development of tris(pentafluorophenyl)borane–catalysed conversion of furans via ring-opening and closing cascade processes to afford silicon-functionalized synthetic chemicals under transition metal-free conditions. The furan ring-opening with hydrosilanes is highly efficient (TON up to 2,000) and atom-economical without forming any byproduct to give rise to $\alpha$-silyloxy-(Z)-alkenyl silanes. Additional equivalents of silane smoothly induce a subsequent $B(C_6F_5)_3$-catalysed cyclization of initially formed olefinic silane compounds to produce anti-(2-alkyl)cyclopropyl silanes, another versatile synthon being potentially applicable in the synthesis of natural products and pharmacophores.

[1] Center for Catalytic Hydrocarbon Functionalizations, Institute for Basic Science (IBS), Daejeon 305-701, South Korea. [2] Department of Chemistry, Korea Advanced Institute of Science & Technology (KAIST), Daejeon 305-701, South Korea. Correspondence and requests for materials should be addressed to S.C. (email: sbchang@kaist.ac.kr).

Production of furans and their subsequent transformations have received a great attention[1,2]. This research activity can be attributed to the relevance of furan derivatives to the renewable biomass. Thermal dehydration of glucose or fructose, the most abundant monosaccharides, in the presence of acid catalysts provides important nonpetroleum chemicals including furfural and 5-hydroxymethylfurfural (HMF)[3–6]. These compounds are considered to be versatile platform chemicals with high potential utility in organic synthesis and industrial applications to give rise to a diverse range of furans and their derivatives. In fact, a number of catalytic transformations of furans or their congeners to valuable chemicals have long been developed in both academia and industry[7–9]. Among the precedents, however, the ring-opening reactions of furans are relatively less highlighted[10–12] when compared with the derivatization of furans maintaining the ring skeleton[13–15]. One notable example of the ring-opening of furans is the production of levulinic acid via an acid-catalysed hydration of HMF (Fig. 1a)[16,17]. Recently, Gordon and coworkers showed that a sequential catalysis involving ring-opening hydrolysis of chain-extended HMF derivatives followed by hydrodeoxygenation using a combination of acid and metal catalyst can produce linear alkanes[18]. In addition, while a borane catalyst system was proved to be applicable in the transformations of carbohydrates, it was also employed for the ring-opening of furan derivatives. For instance, Gagné et al. found that B(C₆F₅)₃ catalyses the deoxygenation of carbohydrate-based polyols with hydrosilanes to give chiral-alcohol synthons[19] or hydrocarbons[20,21] with remarkable chemoselectivity. Njardarson et al. employed B(C₆F₅)₃ catalyst for the reduction of 2,5-dihydrofurans using hydrosilanes to obtain silyl(homo)allyl ethers, which was proposed to form via a hydrosilylative C−O bond cleavage[22]. On the other hand, Ashley et al. reported that hydrogenation of furan derivatives under the frustrated Lewis pairs catalysis using (B(C₆Cl₅)(C₆F₅)₂) delivers reduced tetrahydrofuran compounds[23]. Despite of such progresses on furan transformation reactions, there have been no developments thus far for the selective conversion of furans to ring-opened products bearing a sp³ C−Si bond at ambient conditions.

Described herein is the development of a boron-catalysed cascade silylative transformation of furans (I) involving selective ring-opening and closing processes, thus allowing for the sequential formation of two types of new silylated products of α-silyloxy-(Z)-alkenyl silanes (II) and trans-(2-alkyl)cyclopropyl silanes (IV) (Fig. 1b). In this transformation, several aspects are especially notable: First, while the B(C₆F₅)₃-catalysed hydrosilylation of olefins is known[24], the sp³ C−O and sp² C=C bonds in the initially ring-opened α-silyloxy-(Z)-alkenyl silanes (II) remain unreacted. Second, the subsequent ring-closing of II is readily enabled simply by additional equivalent of hydrosilane leading to silylcyclopropanes (IV) without the need to isolate the ring-opened compounds II. Third, the observed high regio and stereoselectivities in both the initial ring-opening and subsequent cyclopropanation processes are an outcome of kinetic differentiation within the borane catalytic cycle. In addition, both types of products obtained from the present furan conversion are synthetically highly valuable compounds possessing sp³ C−Si bonds[25–27]. Also, a broad range of furan congeners derived from 5-hydroxymethylfurfural are successfully applied for the selective cascade hydrosilylation to yield silicon-functionalized products (Fig. 1c).

## Results

### Discovery of new cascade transformations of furans. When 2-methylfuran was treated with 1.0 equiv. of PhMe₂SiH in the presence of B(C₆F₅)₃ (5.0 mol%) in dichloromethane solvent, a

new product, α-silyloxy-3-pentenyl silane Z-2a, was observed to form in 10 min at −78 °C along with unreacted 1a in a 1:1 ratio (Fig. 2a; see Supplementary Fig. 1). Subsequent addition of 2.0 equivalents of PhMe₂SiH into the above reaction mixture brought about exclusive formation of silylated cyclopropane as a single stereoisomer (anti-3a) in 74% yield over 16 h in addition to (PhMe₂Si)₂O byproduct 4 (Fig. 2a; see Supplementary Fig. 2). These results indicate that the kinetic barrier for the B(C₆F₅)₃-catalysed ring-opening hydrosilylation of 1a to Z-2a is much lower than that of the ring-closing process from Z-2a to anti-3a. Consistent with this interpretation, we were able to confirm that 1a was quantitatively converted to Z-2a with 2.0 equivalents of PhMe₂SiH, while the use of 3.0 equiv. of the silane led to the exclusive formation of anti-3a together with stoichiometric byproduct 4 (Fig. 2b).

To gain more insight into the kinetics of each step of the cascade hydrosilylation processes, we monitored the reaction progress by ¹H NMR spectroscopy (Fig. 2c,d; see Supplementary Fig. 3). A precooled mixture of B(C₆F₅)₃, 1a, and PhMe₂SiH (1:50:200) in CD₂Cl₂ at −70 °C was found to give Z-2a in a quantitative NMR yield over 3.5 h. On further warming up to 25 °C, Z-2a started to subsequently undergo a ring-closing process to afford silylated cyclopropane anti-3a (53% NMR yield in 7 h). These results clearly indicate that the furan transformation proceeds via sequential reductive pathways, which are strongly governed by kinetic factors. Moreover, the initial rates for each process in the hydrosilylation cascade of 2-methylfuran were determined to be $1.14 \times 10^{-4}\,M\,s^{-1}$ (for the ring-opening at −70 °C) and $3.06 \times 10^{-5}\,M\,s^{-1}$ (for the cyclopropanation at 25 °C) (Fig. 2d; Supplementary Fig. 3).

### Substrate scope of the B(C₆F₅)₃-catalysed cascade silylative reduction of furans. With the preliminary results on the 2-methylfuran conversion in hand, we carried out additional optimization studies (Supplementary Tables 1–3) and investigated the substrate scope of this catalysis. 2-Alkyl or 2-arylfuran derivatives employed in this study were easily prepared via the Pd-catalysed Suzuki–Miyaura cross-coupling reaction of (2-furanyl)boronic acid with the corresponding alkyl or aryl halides (Supplementary Methods, GP1). In the present cascade transformation of furans, several notable features were revealed: (i) only equimolar amounts of hydrosilanes are needed to enable the ring-opening process, thus indicating that the conversion is atom-economical; (ii) the reaction is highly stereoselective in that one isomeric products are formed; (iii) the subsequent cyclopropanation process can be carried out without isolation of ring-opened intermediates, α-silyloxy-(Z)-homoallylsilanes; (iv) again, the ring-closing process is highly stereoselective affording anti-products exclusively; and (v) the overall procedure is convenient and easy to scale up with high catalyst turnovers (~2,000).

We were pleased to observe that the optimized conditions were readily applicable to a broad range of furan derivatives to give α-silyloxy-(Z)-homoallylsilanes (Table 1; Conditions A). Substrates with aliphatic substituents on the furan C-2 were all compatible with the B(C₆F₅)₃-catalysed (2.0 mol%) ring-opening process at ambient temperature. Analytically pure α-silyloxy-(Z)-homoallylsilane products (Z-2a~2d) were obtained in high yields and with excellent stereoselectivities (Z/E: >99/1, as measured by crude ¹H NMR experiments). The reaction of aryl-substituted furans proceeded with a similar level of selectivity and efficiency irrespective of the electronic properties of the aryl moiety (Z-2e~2k, Z/E: >99/1, 83–92%). Gratifyingly, potentially reactive functional groups (Z-2l~2n) were tolerated under the reaction conditions, while reactions of the furan substrates

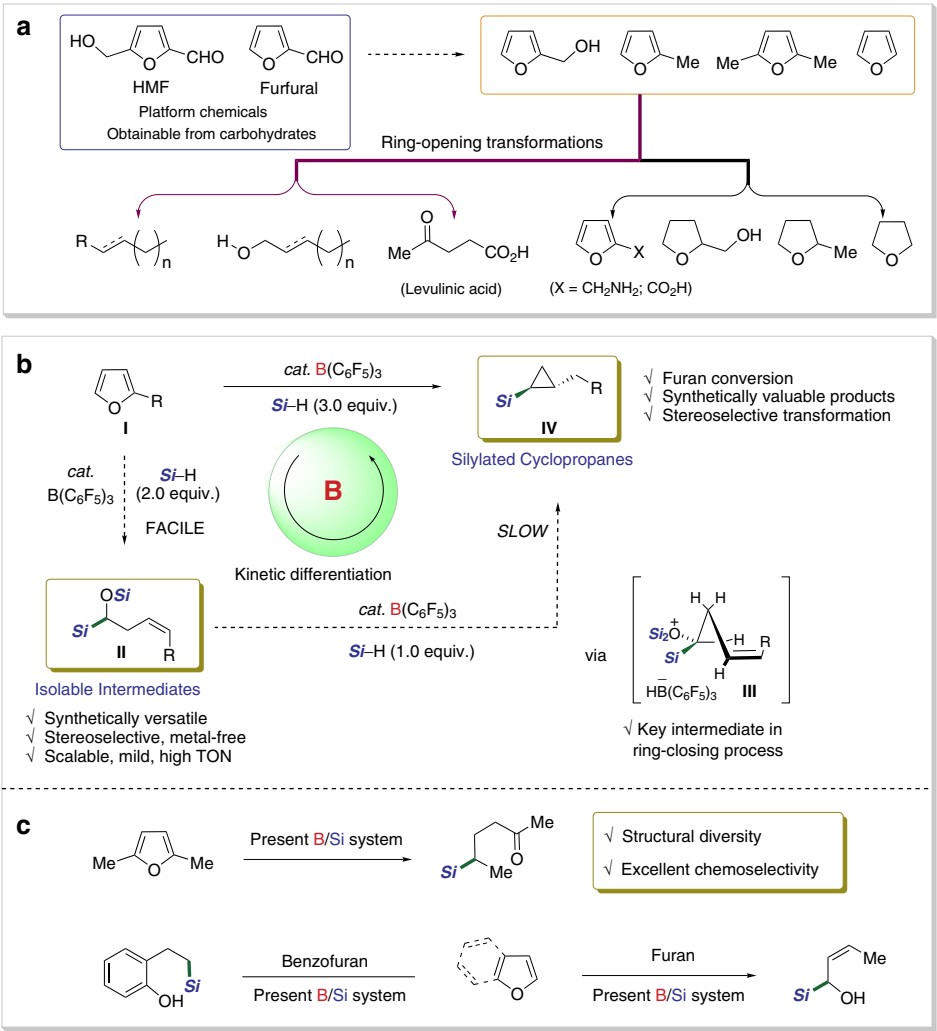

**Figure 1 | Furans from biomass resources and their transformations. (a)** Furan platform chemicals and known post-transformations leading to diverse synthetic valuables. **(b,c)** Present work; Borane-catalysed ring-opening and closing cascades of furans giving rise to synthetically valuable silicon compounds in reaction with hydrosilanes ($Si = SiR'_2R''$).

bearing sterically bulky substituents were also smooth (Z-**2o** ∼ **2q**). In addition, substrates having multiple furan rings (**1r** ∼ **1t**) were readily employed for the current process to afford the corresponding multi-functionalized C-2 or C-3 symmetrical products (**2r** ∼ **2t**). Interestingly, the newly generated multiple double bonds are all in (Z)-form, strongly suggesting that the ring-opening process is highly stereoselective.

We next turned our attention to the formation of silylated cyclopropanes based on the above mechanistic insights (Fig. 2). A brief optimization study led us to establish the one-pot reaction conditions that do not need to isolate α-silyloxy-(Z)-homoallylsilanes compounds. The key in this triple cascade hydrosilylation process was the amounts of hydrosilanes employed: while dimethylphenylsilane was most effective, the use of slightly excessive this silane (4.0 equivalents) was found to be optimal for high product yields. Under these conditions, a range of furan substrates were smoothly converted to the desired 2-alkylcyclopropyl silanes with excellent *anti*-diastereoselectivity in the presence of B(C$_6$F$_5$)$_3$ catalyst (Table 1; Conditions B). *Anti*-diastereoselectivity was confirmed by NMR analyses (Supplementary Figs 114–122). Furans substituted with alkyl groups at the 2-position smoothly underwent the triple hydrosilylation cascade with PhMe$_2$SiH (4.0 equiv.) to afford the corresponding products (*anti*-**3a**, **3b** ∼ **3f**, diastereomeric ratio;

>99/1; the diastereoselectivity was measured by [1]H NMR of the crude reaction mixture). Likewise, 2-arylfurans with varying electronic properties were converted to *anti*-(2-arylmethyl)cyclo-propyl silane in high yields (*anti*-**3g** ∼ **3m**).

Furans bearing multi-substituted phenyl and polyaromatic moieties were also competent substrates for this hydrosilylation cascade (*anti*-**3n** ∼ **3p**). Interestingly, multiple furans connected through a benzene core were smoothly converted to the corresponding products bearing bis- or tris-cyclopropyl groups (**3q** ∼ **3r**, respectively), still displaying an *anti*-stereochemical relationship in each newly generated cyclopropane. It should be mentioned that Stephan, Hashmi, and coworkers recently showed that a stoichiometric reaction of B(C$_6$F$_5$)$_3$ with 1,6-enynes proceeds via initial cyclopropanation and then formal 1,1-carboboration[28]. In addition, Erker *et al.* reported that HB(C$_6$F$_5$)$_2$ can mediate a conversion of allyldimesitylphosphanes to phosphinomethyl-substituted cyclopropane derivatives under frustrated Lewis pair conditions in a stoichiometric manner[29].

**Structural diversity with other furans and benzofurans.** In addition to 2-substituted furans, furan and other regioisomeric derivatives were observed to undergo the selective ring-opening reactions at ambient temperature to give structurally diverse

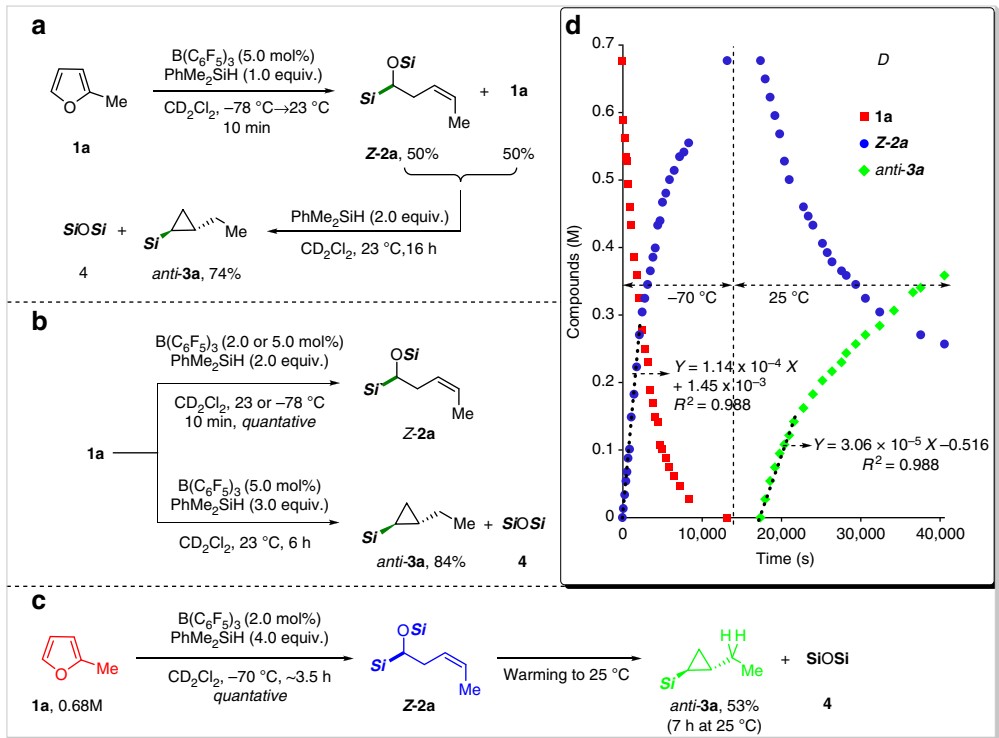

**Figure 2 | Preliminary catalytic reactions and NMR monitoring. (a)** Sequential addition of hydrosilane. (**b**) Control of product distribution by the equivalents of hydrosilane. (**c**) Ring-opening and closing transformations of 2-methylfuran in one flask. (**d**) NMR monitoring of the $B(C_6F_5)_3$-catalyzed cascade conversion of 2-methylfuran with hydrosilane. ($Si = SiMe_2Ph$).

products bearing sp³ C−Si bonds (Table 2). Unsubstituted furan was reacted with $PhMe_2SiH$ under the borane catalysis, leading to (Z)-α-hydroxy allylsilane upon treatment of the reaction mixture with methanol solution of $K_2CO_3$ (Z-**5**, Table 2). 3-Substituted furan was also reactive toward the silylative reduction, furnishing an allylsilane having both silyloxy and aryl moieties at the C1 and C2-positions (Z-**6**). It was interesting to see that disubstituted furan underwent the ring-opening process via slightly modified pathway (see Supplementary Discussion 2). For instance, 2,3-dimethylfuran was reacted under the same conditions to eventually provide γ-silylketone product (**7**) after treating the catalytic reaction mixture with tetrabutylammonium fluoride ($nBu_4NF$) in one pot (**7**, 73% over two steps). In a similar manner, 2,5-dimethylfuran was doubly hydrosilylated to afford the corresponding γ-silylketone (**8**) upon treatment of $nBu_4NF$ (**8**, 75% over two steps). The present procedure was also applicable to the silylative reduction of benzofurans. Reactions of benzofuran or 2-substituted benzofurans with hydrosilane in the presence of borane catalyst selectively proceeded to provide a range of 2-alkyl-substituted phenols (**9~12**) having a sp³ C−Si bond in high yields again after treating the methanol solution of $K_2CO_3$.

**Mechanistic experiments and proposed catalytic pathway.** Based on the present observations and precedents[30,31] regarding the mechanism of $B(C_6F_5)_3$-mediated hydrosilylation, we propose a cascade catalytic pathway leading to the Z-selective homoallylic silane, and subsequently the anti-cyclopropyl silanes, as seen in Fig. 3a using 2-methylfuran as a model substrate. On the in situ generation of a borane-silane adduct **I**, 2-methylfuran **1a** attacks the silylium species to afford an oxonium species **II** that immediately reacts with the borohydride leading to a partially reduced furan intermediate **III** bearing a sp³ C−Si bond next to an oxygen atom. A subsequent hydrosilylation of **III** is suggested to take place through a key intermediate **IV**, at which a selective

borohydride attack at the α-carbon via a nucleophilic vinylic substitution ($S_NV$) pathway leads to α-silyloxy-(Z)-alkenyl silane (**V**)[32] (Fig. 3a (top), **1a** to **V**).

An O-silyl oxonium species **VI** formed by a reaction of **V** with active species **I** is proposed to be involved in the ring-closing process, which will induce the borohydride nucleophilic attack ($S_N2'$-type mechanism) to give the corresponding silylated cyclopropanes with the release of one equivalent of siloxane (Fig. 3a (bottom), **V** to anti-**3a**). With regard to the oxonium species **VI**, two different types of intermediates would be plausible: (i) **VI** that has C1 and C4 substituents (oxonium/silyl and methyl groups, respectively) at the opposite space and (ii) **VI′** having two groups on the same side. The relative stereochemistry (anti-) of two substituents in the resultant cyclopropane product can be reasoned by proposing a nucleophilic substitution of an in situ generated oxonium species **VI** (ref. 20) by the borohydride nucleophile to minimize the steric repulsion between substituents[33].

This mechanistic proposal was supported by a series of experiments, including kinetic and isotopic studies. A hydrosilylation of 2-methylfuran (**1a**) with $PhMe_2SiD$ (2.0 equiv.) under standard conditions led to the exclusive incorporation of deuterium at the two positions to give Z-**2a**-$d_2$ (Fig. 3b). A reaction of 2,3-dihydro-5-methylfuran, **13** with 1.0 equiv. of $PhMe_2SiD$ afforded a product Z-**14**-**d** with a selective deuterium incorporation at the olefinic carbon (Fig. 3c). These results led us to propose a selective and consecutive attack of a borohydride nucleophile following the silylium ion ($R_3Si^+$) (refs 34–38) transfer and the involvement of a partially reduced dihydrofuran intermediate such as **13** during the ring-opening process. Although the proposed vinylic substitution by the borohydride nucleophile is unknown to our best knowledge, an example of an inversion at the alkenyl configuration was previously reported in a reaction of alkylvinyliodonium electrophiles with halide nucleophiles[39].

**Table 1 | B(C₆F₅)₃ catalysed cascade silylative transformation of furans.**

Conditions A; Isolated products yields are presented, in which Z/E was determined by ¹H NMR analysis of the crude reaction mixture.
*Z-2a″: (3,5-dinitro)benzoate derivative from the corresponding alcohol.
†Isolated as a free hydroxyl compound on treating the reaction mixture with K₂CO₃ solution (MeOH): yields over 2 steps.
‡4.0 equiv. of silane was used.
§4.0 Mol% catalyst was used.
‖6.0 Mol% catalyst and 6.0 equiv. of silane were used. (Si = SiMe₂Ph).
Conditions B; In all cases, diastereomeric ratio (dr) was >99/1 determined by ¹H NMR analysis of the crude reaction mixture. (Si = SiMe₂Ph). ¶Ph₂SiH₂ (3.0 equiv.) was used.
#Conducted at 40 °C.
**10.0 Mol% catalyst and 8.0 equiv. of silane were used.
††20.0 Mol% catalyst and 12.0 equiv. of silane were used.

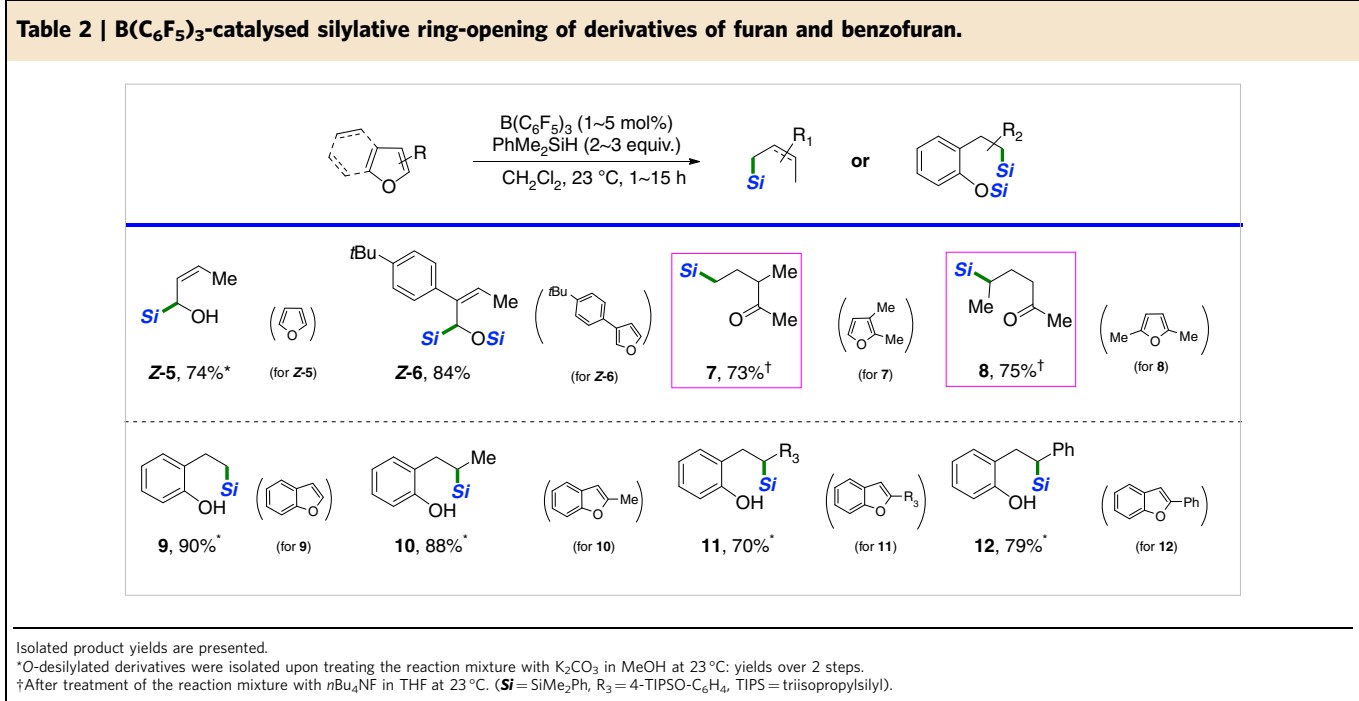

**Table 2 | B(C₆F₅)₃-catalysed silylative ring-opening of derivatives of furan and benzofuran.**

Isolated product yields are presented.
*O-desilylated derivatives were isolated upon treating the reaction mixture with K₂CO₃ in MeOH at 23 °C: yields over 2 steps.
†After treatment of the reaction mixture with nBu₄NF in THF at 23 °C. (**Si** = SiMe₂Ph, R₃ = 4-TIPSO-C₆H₄, TIPS = triisopropylsilyl).

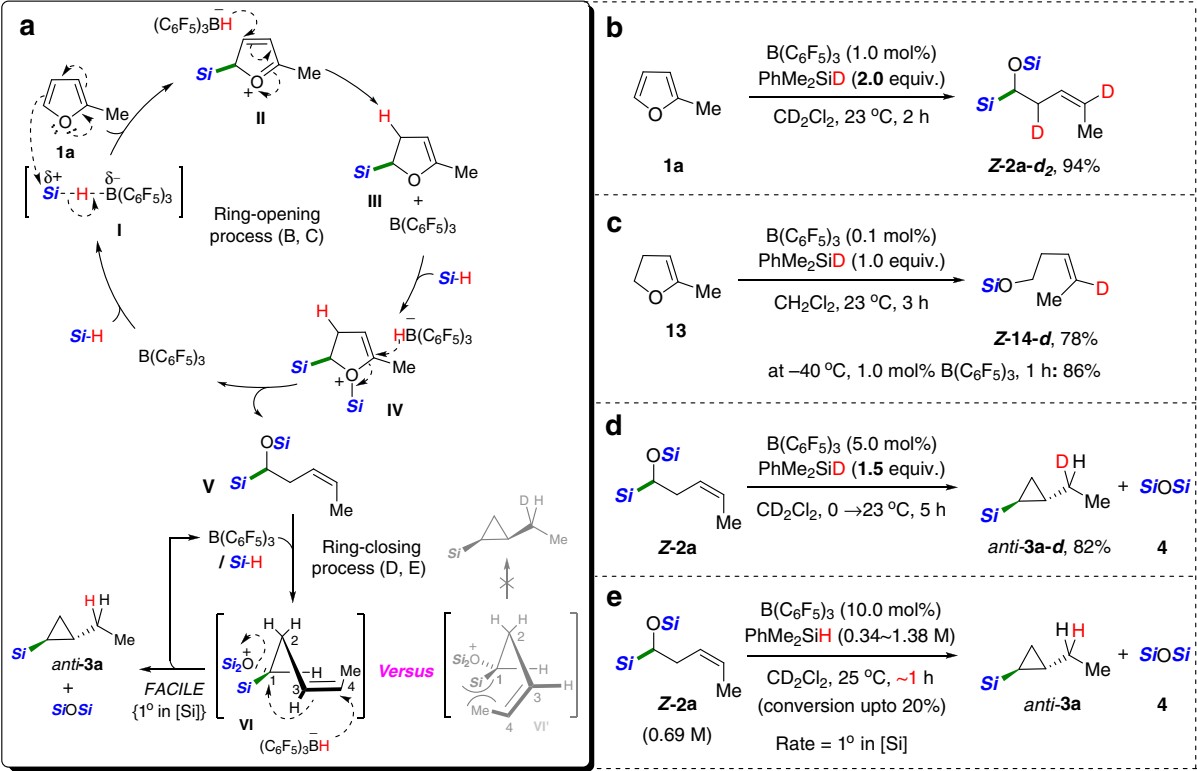

**Figure 3 | Mechanistic experiments and proposed catalytic pathway.** (**a**) Proposed pathway of the B(C₆F₅)₃-catalysed silylative ring-opening of furans and subsequent cyclopropanation. For simplicity, hydrosilane is shown as **Si**-H/D (**Si** = SiMe₂Ph). (**b**) Deuterium labelling experiment. (**c**) A model reaction with 2,3-dihydro-5-methylfuran. (**d**) A test ring-closing reaction of Z-**2a** with 1.5 equiv. of deuterated hydrosilane. (**e**) Rate-order assessment of silane based on initial rates in a range of initial silane concentrations.

As expected, the subsequent ring-closing process from **V** to the corresponding silylated cyclopropanes was highly stereoselective as proved by an isotope experiment. Indeed, when α-silyloxy-(Z)-alkenyl silane (Z-**2a**) was allowed to react with PhMe₂SiD (1.5 equivalents), cyclized product (anti-**3a-d**) was obtained in 82% yield as a single isomer with the exclusive incorporation of deuterium at the α-ethyl position (Fig. 3d). A stoichiometric amount of siloxane was also confirmed to be generated during the cyclopropanation process. An initial-rate study for the cyclopropanation of **V** under the B(C₆F₅)₃-catalysed hydrosilylation

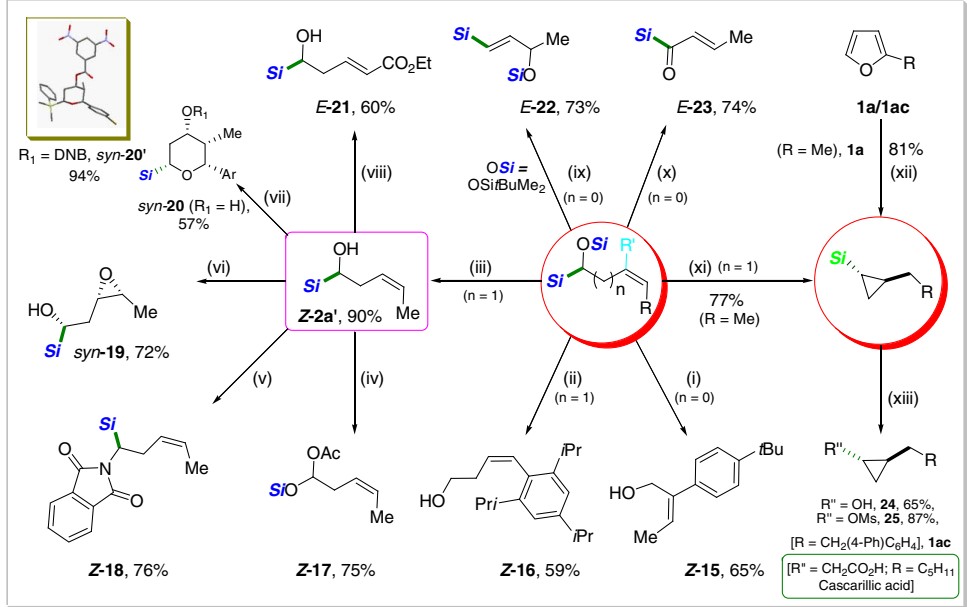

**Figure 4 | Enrichment and elaboration of products.** (i) CsF (5 equiv.), 23 °C, 8 h, DMF, (R′ = (tBu)-C₆H₄, R = Me). (ii) TBAF (3 equiv., 1 M in THF), 23 °C, 15 h, (R′ = H, R = (2,4,6-triisopropyl)C₆H₂). (iii) K₂CO₃ (2 equiv.), MeOH, 0→23 °C, 4 h, (R′ = H, R = Me). (iv) Pb(OAc)₄ (1.1 equiv.), C₆H₆, 0→23 °C, 2 h. (v) PPh₃ (1.3 equiv.), Diethyl azodicarboxylate (1.3 equiv.), Phthalimide (1.3 equiv.), THF, 0→23 °C, 20 h. (vi) VO(acac)₂ (5 mol%), TBHP (2.2 equiv.), CH₂Cl₂, −10→0 °C, 5 h. (vii) p-Br-C₆H₄CHO (1.2 equiv.), TFA (2.5 equiv.), CH₂Cl₂ (3 ml), −20 °C, 3 h; DNB = 3,5-dinitrobenzoyl. (viii) Ethyl acrylate (10 equiv.), Grubbs-II (5 mol%), 50 °C, 15 h. (ix) BF₃•Et₂O (1.1 equiv.), CH₂Cl₂ (0.14 M), −78 °C, 5 h, (R′ = H, R = Me). (x) (COCl)₂ (2.0 equiv.), DMSO (4.0 equiv.), Et₃N (5.0 equiv.), −78 °C, 4 h, (R′ = H, R = Me). (xi) B(C₆F₅)₃ (3.0 mol%), PhMe₂SiH (1.5 equiv.), CH₂Cl₂, 0→23 °C, 12 h, (R′ = H, R = Me). (xii) 2-Methylfuran (**1a**, 1.0 equiv.), B(C₆F₅)₃ (5.0 mol%), Ph₂SiH₂ (3.0 equiv.), CH₂Cl₂, 0→23 °C, 8 h. (xiii) (a) **1ac** (1.0 equiv.), B(C₆F₅)₃ (5.0 mol%), Ph₂SiH₂ (3.0 equiv.), CH₂Cl₂, 0→23 °C, 10 h; (b) H₂O₂ (20 equiv.), KF (10 equiv.), KHCO₃ (10 equiv.), THF/MeOH (1/1), 23 °C, 16 h. (**Si** = SiMe₂Ph, **Si** = SiPh₂H/SiMe₂Ph).

conditions clearly showed that the reaction was first-order in silane concentration (Fig. 3e; see Supplementary Figs 4 and 5), leading us to propose that the cyclization path involves an O-silylated oxonium ion possessing a borohydride anion[40].

**Synthetic applications.** The synthetic utility of the products obtained was demonstrated to be very broad, including a wide range of simple and convenient organic transformations (Fig. 4). α-Silyloxy-(Z)-alkenyl silanes were easily desilylated to give allylic- or homoallylic alcohols with retention of double bond stereochemistry (Z-15 and Z-16, respectively). α-Hydroxy homoallylic silane (Z-2a′), accessed through the O-desilylation of α-silyloxy-(Z)-homoallylic silane, turned out to be also synthetically versatile. α-Silyloxy acetate (Z-17) was readily produced by a radical Brook rearrangement in good yield[41]. Moreover, Z-2a′ was smoothly converted to (Z)-α-amino silane derivative (Z-18) under the Mitsunobu conditions leading to C−N bond formation. Epoxide (19) was obtained syn-selectively via the Sharpless directed epoxidation of Z-2a′ (ref. 42), thus proving that α-silylalcohol works as an effective directing group. An efficient Prins cyclization[43] of Z-2a′ was achieved to furnish a tetrasubstituted pyran ring (syn-20) with complete stereocontrol. The observed syn-stereochemistry was unambiguously confirmed by NMR and X-ray diffraction analyses. Cross metathesis of Z-2a′ with ethyl acrylate was carried out stereoselectively with Grubbs II catalyst to give E-21. Lewis acid-promoted allylic rearrangement[44] of α-silyloxy-(Z)-allylic silane bearing tert-butyldimethylsilyloxy group Z-5″ (tert-butyldimethylsilyl derivative of Z-5, see Supplementary Methods) proceeded smoothly to afford silyloxy vinylic silane (E-22), another versatile synthetic building block. Acylsilane (E-23), a valuable organosilicon reagents in organic synthesis[45], was readily obtained.

Two preparative procedures of cyclopropyl silanes were mild and efficient on a large scale (Supplementary Methods): (i) a stepwise route via α-silyloxy homoallylic silane (Z-2a, procedure "xi" in 77%), and (ii) a direct conversion of furan (procedure "xii" in 81% yield). The subsequent oxidation of the C−Si bond under the Tamao–Fleming conditions[46] proceeded with retention of stereochemistry to furnish the versatile synthons, cyclopropanol (24) and its methansulfonyl derivative (25), which are broadly utilized in synthetic and medicinal chemistry[47]. For instance, cascarillic acid[48], a natural product derived from the bark of the medicinal shrub *Croton eluteria*, can be envisioned to be accessible based on our current procedure.

In summary, we have developed the tris(pentafluorophenyl)-borane-catalysed cascade transformation of furans that are readily available from renewable biomass resources to obtain synthetically valuable silicon-functionalized products such as α-silyloxy-(Z)-alkenyl silanes and anti-cyclopropyl silanes. Simply by varying the stoichiometry of employed hydrosilanes in presence of B(C₆F₅)₃ catalyst, the product distribution could completely be controlled with high efficiency (TON up to 2,000) and excellent stereoselectivity. The transformation does not require transition metal catalysts, proceeds efficiently on large scale, and is broadly applicable to various types of furans and their derivatives, bringing about a significant structural diversity in the product obtained. A proposed mechanistic pathway involves a series of hydrosilylation cascades, containing a ring-opening and subsequent SN2′-type ring-closing process, both mediated by a B(C₆F₅)₃ catalyst. The synthetic utility of obtained silicon-functionalized products was demonstrated in a range of post-transformations. It is anticipated that this study will stimulate future developments in the transformations of biomass-derived platform chemicals to synthetic valuables.

## Methods

**General procedure for the silylative ring opening reaction (conditions A, GP1).** In a flame-dried flask bearing a stirring bar, $B(C_6F_5)_3$ (0.01 ~ 0.02 mmol, 2.0 mol%) was dissolved in $CH_2Cl_2$ (0.4 ~ 0.8 ml). Silane (1.025 ~ 2.050 mmol) was added, and the solution was shaken shortly to make it homogeneous. The corresponding furan derivative (**1a** ~ **1t**, 0.50 ~ 1.0 mmol) was then added and the reaction mixture was stirred at 23 °C for the indicated time (1 ~ 15 h). After quenching the reaction mixture with $Et_3N$ (10.0 ~ 20.0 mol%), the crude reaction mixture was concentrated under reduced pressure and then purified by flash column chromatography on silica gel (using either hexane only or a mixture of hexane/ethyl acetate) to afford the desired products (Z-**2a** ~ Z-**2t**, in all cases Z/E > 99/1).

**General procedure for the cyclopropanation reaction (conditions B, GP2).** In a flame-dried flask bearing a stirring bar, $B(C_6F_5)_3$ (0.025 mmol, 5.0 mol%) was dissolved in $CH_2Cl_2$ (0.2 ml). Silane (1.5 ~ 2.0 mmol) was added, and the solution was shaken shortly to make it homogeneous. The corresponding furan derivative (**1a** ~ **1t**, 0.50 mmol) was then added at 0 °C and the reaction mixture was stirred at 23 °C for the indicated time (4 ~ 22 h). After quenching the reaction mixture with $Et_3N$ (10.0 ~ 20.0 mol%), the crude reaction mixture was concentrated under reduced pressure and then purified by flash column chromatography on silica gel (using either hexane or a mixture of hexane and ethyl acetate) to afford the desired products (anti, **3a** ~ **3r**, in all cases dr > 99/1).

**Data availability.** The authors declare that the data supporting of the findings of this study are available within the article and Supplementary Information files. For the experimental procedures and spectroscopic and physical data of compounds, see Supplementary Methods. For NMR analysis of the compounds in this article, see Supplementary Figs 6–187. The CCDC 1505482 (Z-**2a″**) and CCDC 1505484 (syn-**20′**) contain the supplementary crystallographic data for this paper (Supplementary Tables 4 and 5). These data can be obtained free of charge from The Cambridge Crystallographic Data Centre via http://www.ccdc.cam.ac.uk/data_request/cif. All other data are available from the authors on reasonable request.

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

## Acknowledgements

This article is dedicated to Professor Naoto Chatani on the occasion of his 60th birthday. This research was supported by the Institute for Basic Science (IBS-R10-D1) in Korea. We are thankful to Prof. Mu-Hyun Baik for his inspiring discussions and critical reading.

## Author contributions

C.K.H., N.G., and S.P. carried out the experiments. C.K.H., N.G. and S.C. designed and directed this project. S.P. and S.C. wrote the manuscript. All authors analysed the data, discussed the results and commented on the manuscript.

## Additional information

**Competing financial interests:** The authors declare no competing financial interests.

**Publisher's note**: 

