## [Peer Review File · Nature Communications]

Transferred manuscripts:

Reviewer #1 (Remarks to the Author):

There are some issues with the structure refinement of 'anti-3b'.

Namely, the group containing atoms C9, C10, C11 and C12 should be refined as disordered over at least two sets of sites. The disorder is indicated by very large anisotropic displacement parameters for the atoms named above.

Additionally, using the anisotropic model instead of the disorder model has created some distortion of the bond lengths from the expected values. The disorder appears to be the result of some rotation about the C8--C6 bond. I highly recommend that this be attended to before the crystal structure is acceptable.

There are also some A-Alerts for 'anti-3b'. The authors should include validation reports in the CIF file to address the following:

THETM01_ALERT_3_A The value of $\sin(\theta_{\max})/\lambda$ is less than 0.550
Calculated $\sin(\theta_{\max})/\lambda = 0.5127$

PLAT029_ALERT_3_A $\text{diffn_measured_fraction_theta_full}$ value Low . 0.903 Note

In addition, for structure 'Z 2n' there is a discrepancy, in the CIF between the reported formula and that calculated from the contents of the CIF.

This should also be addressed.

■ Detailed Responses to the Reviewer #1's Comments:

Q-1) "There are some issues with the structure refinement of 'anti-3b'. Namely, the group containing atoms C9, C10, C11 and C12 should be refined as disordered over at least two sets of sites. The disorder is indicated by very large anisotropic displacement parameters for the atoms named above. Additionally, using the anisotropic model instead of the disorder model has created some distortion of the bond lengths from the expected values. The disorder appears to be the result of some rotation about the C8-C6 bond. I highly recommend that this be attended to before the crystal structure is acceptable. There are also some A-Alerts for 'anti-3b'. The authors should include validation reports in the CIF file to address the following: THETM01_ALERT_3_A The value of $\sin(\theta_{max})/\lambda$ is less than 0.550 Calculated $\sin(\theta_{max})/\lambda = 0.5127$ PLAT029_ALERT_3_A $\text{diffn_measured_fraction_theta_full}$ value Low . 0.903 Note."

Response:

Although tired hard to better quality of X-ray data for the indicated compound (*anti-3b*) since our initial submission in August, unfortunately no better results we obtained still showing Alerts A and B. However, since we have solid NMR interpretation to assign the *anti*-diastereoselectivity of disubstituted cyclopropane products (the same conclusion obtained from the low quality X-ray) through the NMR analysis of one representative product, *anti-3m*, we decided to delete the X-ray data of *anti-3b*. Instead, new NMR analysis is now included the revised supporting information (see SI, page S153-S156). As we believe that our NMR data is good enough to assign the stereochemistry as well as cyclopropane ring formation, we delete all crystal data of *anti-3b* (page 5, Table 1 in the original manuscript of highlighted version).

As a result, we have made a few changes regarding the crystallographic data as follows:

- Characterization data, spectral data, and crystallographic data of *anti-3b* and *anti-3b* are deleted from the original S.I. as highlighted in yellow (page S28-S29, S134-S136, S242-S248 respectively)

Q-2) "In addition, for structure 'Z-2n' there is a discrepancy, in the CIF between the reported formula and that calculated from the contents of the CIF. This should also be addressed."

Response:

In the same line with the above answer, crystallographic data of *Z-2n* is now deleted in the revised text and supporting information. Instead, the presence of a crystal structure of *Z-2a* is now newly indicated in the revised text (see Table 1 and page 5) to confirm the stereochemistry of ring-opened products.

As a result, we revised and changes a few things as follows:

- Characterization data, spectral data, and crystallographic data of *Z-2n* is deleted from the original S.I. highlighted version (page S24, S116-S117, and S233-S241 respectively)